# Associations of Calcium Intake and Calcium from Various Sources with Blood Lipids in a Population of Older Women and Men with High Calcium Intake

**DOI:** 10.3390/nu14061314

**Published:** 2022-03-21

**Authors:** Maria Papageorgiou, Fanny Merminod, Serge Ferrari, René Rizzoli, Emmanuel Biver

**Affiliations:** Division of Bone Diseases, Geneva University Hospitals and Faculty of Medicine, University of Geneva, 1205 Geneva, Switzerland; maria.papageorgiou@unige.ch (M.P.); fanny.merminod@hcuge.ch (F.M.); serge.ferrari@unige.ch (S.F.); rene.rizzoli@unige.ch (R.R.)

**Keywords:** calcium intake, calcium supplements, dairy consumption, fermented dairy products, low-fat dairy products, dyslipidemia, cholesterol, triglycerides

## Abstract

Promoting calcium intake is a cornerstone for osteoporosis management. Some individuals limit dairy product consumption, a major calcium source, due to their high content in saturated fats and their perceived negative impact on lipid profiles. This study explored the associations of calcium from various sources with blood lipids in community-dwelling elderly (*n* = 717) from the GERICO cohort. Dietary calcium intake was assessed at several timepoints using a validated food frequency questionnaire (FFQ) and calcium supplement use was recorded. Blood lipids were treated as categorical variables to distinguish those with normal and abnormal levels. Increasing total calcium intake was associated with lower risks for high total cholesterol (*p* = 0.038) and triglycerides (*p* = 0.007), and low HDL-cholesterol (*p* = 0.010). Dairy calcium (*p* = 0.031), especially calcium from milk (*p* = 0.044) and milk-based desserts (*p* = 0.039), i.e., low-fat (*p* = 0.022) and non-fermented (*p* = 0.005) dairy products, were associated with a lower risk of high total cholesterol. Greater calcium intakes from total dairies (*p* = 0.020), milk (*p* = 0.020) and non-fermented dairies (*p* = 0.027) were associated with a lower risk of hypertriglyceridemia. No association was observed between calcium from non-dairy sources, cheese or high-fat dairies and blood lipids. Increasing calcium through supplements was associated with lower risks for hypertriglyceridemia (*p* = 0.022) and low HDL-cholesterol (*p* = 0.001), but not after adjustments. Our results suggest that higher calcium intakes from dietary sources or supplements are not adversely associated with blood lipids in the elderly, whilst total, and particularly low-fat, dairy products are valuable calcium sources potentially related to favorable lipid profiles.

## 1. Introduction

Adequate calcium intake is critical for the attainment of peak bone mass and bone mineralization, but also for osteoporosis prevention and treatment [1,2,3]. In many countries, dairy products make important contributions to total calcium provision [4,5]. This is largely because dairy products are rich in absorbable calcium and other related nutrients (e.g., proteins, vitamin D, magnesium) and readily available at a relatively low cost [6]. Many dietary guidelines recommend the daily consumption of 2–4 portions of dairy products, which would cover most of the calcium requirements at population level [7,8,9]. Nevertheless, some individuals, including some postmenopausal women and elderly at risk of bone fragility, may refrain from or limit their dairy product consumption, due to their high content in saturated fatty acids (SFAs) and their potential negative impact on lipid profiles and cardiovascular disease (CVD) risk [8,10,11,12,13].

Interestingly, several lines of evidence suggest that the purported negative effects of SFAs could be offset by the favorable effects of some dairy food matrices [12,13,14,15,16]. Indeed, recent meta-analyses found either no association or a lower CVD risk with increasing consumption of total dairy products [17,18,19]. Similarly, although SFAs have been traditionally thought to increase low-density lipoprotein cholesterol (LDL-c) levels, several meta-analyses do not support significant effects of dairy intake on total or LDL-cholesterol (reviewed in [17]), with these findings being consistent for dairy products with different fat content (i.e., regular- or low-fat dairy products) [20]. Further supporting the complex interactions of nutrients/bioactive ingredients within different food matrices, fermented dairy products and especially yogurt, which are sources of probiotics, have been shown to exert some benefits on blood lipids and other metabolic risk factors (for relevant reviews see [16,21,22,23,24,25]). One hypothesis to explain these observations is that calcium in dairy products may influence the intestinal absorption and metabolism of lipids [26,27], and thus, it would be interesting to investigate whether calcium from different dairy sources has differential effects on lipid parameters.

Conversely, populations who rely less on or avoid dairy products derive calcium from other dietary, non-dairy sources (e.g., plant sources and mineral waters); nonetheless, they often fail to achieve adequate intakes, and calcium from some of these sources is less bioavailable [6,28]. It remains unknown whether calcium from non-dairy sources influences blood lipid parameters in a similar or different way to dairy sources (for a given amount of total calcium intake). Finally, calcium supplements are another way to meet calcium needs. They are commonly prescribed by healthcare professionals to postmenopausal women and the elderly with low dietary calcium intakes at risk of bone fragility. Given that calcium supplements are also available over the counter, they may also be regularly consumed by individuals with adequate dietary calcium intakes, resulting in high calcium intakes exceeding recommendations. Although calcium supplements have been shown to exert overall beneficial effects on cardiometabolic risk factors including blood lipids [29,30,31,32], it is less clear whether these effects depend on total calcium intake and calcium intakes from different foods.

We aimed to investigate the associations of total calcium intake and calcium from different sources with blood lipids in community-dwelling elderly with high calcium intakes. Specifically, by building a number of different models that considered the total amount of calcium and adjusting each one for other sources of calcium, we aimed to tease out the differences in calcium intake from diet vs. supplements, dairy vs. non-dairy sources and by dairy subtype, fermentation status and fat content.

## 2. Materials and Methods

### 2.1. Study Population

The present study is a cross-sectional analysis of data from the Geneva Retirees Cohort (GERICO), the design of which has been described in detail elsewhere [33]. Briefly, GERICO was designed to evaluate genetic, musculoskeletal and lifestyle factors including dietary habits in association with bone microstructure and fracture risk in recently retired men and women residing in Geneva and the wider Geneva area in Switzerland. Healthy, community-dwelling individuals aged 63–67 years old were recruited around the time of their retirement during the period 2008–2011 by advertisements placed in the local press, Geneva University Hospitals, or workplaces. Participants were excluded if they had major medical conditions known to influence bone health. In addition to a baseline visit, participants were invited to attend 2 follow-up visits at intervals of ~3 years during the periods 2012–2014 (1st follow-up) and 2015–2018 (2nd follow-up), during which musculoskeletal health traits, metabolic parameters and dietary and other lifestyle factors were assessed. The present analysis was performed using blood lipid data assessed at the 2nd follow-up visit (*n* = 717). The protocol of the GERICO was granted approval by the Geneva University Hospitals’ Ethics Committee and participants signed an informed consent prior to any testing. The study is registered as a clinical trial (http://www.isrctn.com/, No: ISRCTN11865958, approved on the 21 July 2016).

### 2.2. Assessment of Calcium from Different Sources

A validated food frequency questionnaire (FFQ) containing 25 items and designed to assess calcium and protein intakes was used to assess habitual dietary intake, including dairy product consumption over the preceding year at all study visits (adapted from [34]). The questionnaire was administered by trained research staff during a face-to-face interview with the participants at all study visits. Individuals indicated the frequency they consumed specific food items and serving sizes were estimated with the help of pictures used in the large multi-center survey (SU.VI.MAX). The responses to individual items were converted into mean daily consumption (grams per day) by multiplying the typical portion sizes (grams) by the consumption frequency for each food and making the appropriate division for the period assessed to obtain daily consumption. Estimates of calcium, energy and protein intakes were based on the Centre d’Information sur la Qualité des Aliments (Information Center on Food Quality, CIQUAL) food composition tables (Ed. Tec&Doc, Lavoisier and INRA, Maison Alfort, France, 1995). Consumption of dairy products was categorized by subtype into milk (including total, low-fat and whole-fat), yogurt (including total, low-fat and whole-fat yogurt, plain or with fruits), and cheese (fresh cheese such as cottage cheese, fromage frais, “petit suisse”, soft cheese such as Camembert and Tomme and hard cheese such as parmesan, fondue, raclette, grated cheese), and milk-based desserts (e.g., flan, curd, puddings, custard) (Table 1). As fermented dairy products, we considered all types of yogurt and cheese, while milk and milk-based desserts were classified as non-fermented dairies. Low-fat dairy products were defined as milk, yogurt, fresh cheese or milk-based desserts with a fat concentration <20 g/100 g. High-fat dairy was defined as dairy products with a fat concentration ≥20 g/100 g [35]. Calcium from non-dairy sources was considered the sum of calcium from vegetables, legumes, cereals, tofu, meat/fish and their products and mineral waters. The mean values of total calcium intake and calcium from the aforementioned sources were calculated using data from all available dietary assessments (baseline + one or two follow-up visits) to minimize within-subject variation in the diet and capture long-term dietary habits over the follow-up period (6.1 ± 1 years). At the second follow-up visit, participants were also asked to report if they were taking any calcium supplements and provide information about the type and the dosage of the supplement.

### 2.3. Blood Lipids Profile and Related Factors

Blood samples were drawn in the morning following an overnight fast. Serum and plasma were isolated and stored at −80 °C until batch analyses. Total cholesterol, high-density lipoprotein cholesterol (HDL-c) and triglycerides (TG) were measured in plasma samples available at the last follow-up visit using ultraviolet (UV)-visible spectroscopy. LDL-c was calculated using the Friedewald formula [36]. According to the recently updated guidelines of the European Society of Cardiology (ESC), those with LDL-c levels ≥ 2.6 mmol/L were identified as having high LDL-c [37]. The cut-offs of the International Diabetes Federation (IDF) for metabolic syndrome were used to classify those with low HDL-c (<1.3 mmol/L, the cut-off for women was used given the greater proportion of women in our study) and high TG levels (≥1.7 mmol/L) [38]. Given that there is no specific goal for total cholesterol, a cut-off of ≥6.5 mmol/L was used to identify those with levels according to standard local laboratory references. Participants were asked to report any CVDs and the use of anti-hyperlipidemic treatments in an interview with a medical doctor. CVD risk for fatal and non-fatal (myocardial infarction, stroke) CVDs in populations at low CVD risk (Switzerland) was estimated according to the recent guidelines of ESC [37].

### 2.4. Assessment of Covariates and Secondary Outcomes

Height (measured using a Harpenden Stadiometer [Holtain Ltd., Crosswell, United Kingdom]) and weight (measured to the nearest 0.1 kg using standard electronic scales) were used to calculate body mass index [BMI] (kg/m^2^). Participants were asked questions about any other medication used (including the use of anti-osteoporotic, anti-diabetic and antihypertensive treatments) and lifestyle factors (current smoking [yes/no], alcohol consumption [g/week]) in an interview with a medical doctor. Physical activity (mean of all visits) was assessed by a face-to-face questionnaire that uses an inventory of 45 activities to estimate the time spent on usual walking, cycling, stair climbing, organized sports, and recreational activity over the past 12 months. The collected data were converted to physical activity energy expenditure (kilocalories per day) using validated formulas [39]. Established clinical risk factors of fractures were assessed as part of the evaluation of the 10-year probability of fracture using the Fracture Risk Assessment (FRAX) tool (www.shef.ac.uk/FRAX/, last accessed on the 15 June 2018) [40]. Areal BMD (aBMD) at lumbar spine, proximal femur and total radius was assessed using dual-energy X-ray absorptiometry (DXA) (Hologic QDR Discovery instrument Hologic Inc., Waltham, MA, USA). Serum β-carboxyterminal cross-linked telopeptide of type I collagen (β-CTX), total 25-hydroxyvitamin D and parathyroid hormone (PTH) were assessed at the baseline visit of the cohort on a Cobas-6000 analyzer using Elecsys reagents (Roche Diagnostics, Rotkreuz, Switzerland). Creatinine was determined by UV absorption spectrophotometry on a routine procedure analyzer.

### 2.5. Statistical Analysis

Baseline characteristics are expressed as means ± SD for continuous variables or as percentages for categorical variables. Data distributions were checked for normality using the Shapiro–Francia W test and skewness/kurtosis tests. As most parameters had non-Gaussian distributions and were mostly not normalized using simple mathematical transformations, the Huber–White robust sandwich estimator of standard errors to account for the lack of normality of our data. The associations of calcium and calcium sources at baseline with bone-related markers (PTH and β-CTX) were assessed using multivariate linear regression models adjusted for potential confounders. Blood lipids were treated as categorical variables to distinguish those with normal and abnormal levels. We fitted multivariate logistic regression models to examine the likelihood of having high total and LDL-c, low HDL-c and high TG according to total calcium intake (Model 1) and calcium sources (Models 2–6). Specifically, analyses were performed for dietary calcium and calcium from supplements (Model 2), dietary calcium from dairy and non-dairy sources (Model 3) and dairy calcium by subtype (Model 4), fermentation status (Model 5) and fat content (Model 6). For an easier clinical interpretation of our data, the results represent modified odds for having abnormal lipids for each increase of 300 mg calcium, which is approximately the quantity included in one serving of dairy products [6]. All models considered the total amount of calcium consumed and were adjusted for other sources of calcium. Further adjustments were made for age, sex, smoking status, alcohol intake, physical activity energy expenditure, dietary energy intake and drugs that may affect blood lipids or dietary patterns. A series of sensitivity analyses was also conducted to test the robustness of our results. Given that individuals may alter their diet, such as avoiding high-fat dairy products or opting for low-fat and/or fermented dairy products as a result of disease, treatment, or enhancement of health awareness, we tested the associations between calcium from these sources and abnormal blood lipids in subgroups by sex, CVD risk (low/moderate vs. high/very high), osteoporosis status (no vs. yes), previous major osteoporotic fracture (no vs. yes) and use of statins and osteoporosis treatment (no vs. yes). We evaluated a potential modification effect by adding an interaction term of calcium from fermented/high-/low-fat dairy products and the stratifying variable. The level of significance for all statistical tests was *p* < 0.05. All statistical analyses were performed with STATA software, version 14.0 (StataCorp LP, College Station, TX, USA).

## 3. Results

### 3.1. Subjects’ Characteristics

The study population consisted of 717 participants (80% women) with an age (mean ± SD) of 71 ± 2 years and a mean BMI of 25.5 ± 4.4 kg/m^2^ (Table 2). Mean total Ca intake was 1527 ± 527 mg/day. Most of the participants (84%) were meeting Ca recommendations in Switzerland (1000 mg/d) and only 4% of them were consuming <700 mg Ca/day [41]. The largest contributors to total Ca intake were dairy products (46%) with cheese, yogurt, and milk representing 26, 11 and 7% of total intake, respectively (Table 1). Forty percent of the participants consumed ≥3 servings of dairy products daily. Almost one in two participants (46%) were taking Ca supplements, which were accounting for 18% of total daily Ca intake. The remaining 37% of daily Ca intake originated from non-dairy sources including other animal non-dairy products, plant-based sources (e.g., vegetables, legumes, cereals and tofu), and mineral waters rich in calcium. With regards to bone health, the prevalence rates of osteoporosis (at least one BMD T-score ≤ −2.5 SDs at the spine or hip) and low-trauma fractures were 21% and 26%, whilst 106 individuals (15%) were receiving antiresorptive medications. Dietary Ca intake, in particular Ca from dairy products, was associated with lower PTH and β-CTX (a marker of bone resorption) levels (Table 3). One hundred and seventy-eight individuals (25%) self-reported to have dyslipidemia and/or taking statins or other treatments for dyslipidemia. Participants’ mean levels of blood lipids are presented in Table 2. The proportion of participants with high total cholesterol, high LDL-c, low HDL-c and high TG were 21, 80, 16 and 14%, respectively. Based on the SCORE2/SCORE2-OP charts, 36%, 54% and 10% of the participants were at low/moderate, high and very high 10-year CVD risk. Due to the low number of participants categorized as having very high CVD risk, in subsequent analyses, we pooled together those at high and very high risk (64% of the population). Individuals at low/moderate CVD risk had a higher total Ca intake compared to those at high/very high risk (Ca intake: 1615 ± 522 mg/day vs. 1477 ± 523 mg/day, respectively; *p* < 0.001).

### 3.2. Associations between Total Calcium Intake and Lipid Profile

Table 4 shows the univariate and adjusted multivariate logistic regression models for high total cholesterol and TG levels according to total Ca intake and Ca sources, whilst Table 5 further details these associations for having abnormal sub-fractions of total cholesterol, namely high LDL-c and low HDL-c levels.

We first explored the associations of total Ca intake with abnormal blood lipids (Model 1). Increasing total Ca intake was associated with a lower risk of having high levels of total cholesterol [OR (95% CI) 0.90 (0.81, 0.99), *p* = 0.038) and TG (OR 0.83 (0.73, 0.95), *p* = 0.007], and low levels of HDL-c [OR 0.85 (0.75, 0.96), *p* = 0.010]. These associations were attenuated in models adjusted for pre-specified factors including age, sex, weight, height, smoking status, alcohol intake, dietary energy intake, physical activity energy expenditure and use of lipid-lowering, antihypertensive and antidiabetic drugs. No significant association was observed between total calcium intake and LDL-c.

### 3.3. Associations between Calcium Sources and Lipid Profile

We further explored whether the inverse associations between total Ca intake and abnormal blood lipids were dependent on calcium sources (dietary Ca vs. Ca supplements, Model 2; dairy vs. non-dairy Ca, Model 3). The inverse association between total calcium intake and high total cholesterol levels was mainly attributable to dietary Ca [OR 0.83 (0.71, 0.96), *p* = 0.014), while for high TG levels we observed favorable associations with both dietary Ca [OR 0.85 (0.71, 1.02), *p* = 0.081] and Ca from supplements [OR 0.82 (0.69, 0.97), *p* = 0.022] (Model 2 in Table 4). These associations were also attenuated after adjustments for potential confounding factors. Increasing Ca from supplements was associated with a lower risk of low HDL-c in univariate models, but not after adjustments. Concerning the sources of dietary Ca, significant negative associations were observed between Ca from dairy products and elevated total cholesterol and TG levels, which largely remained significant after adjusting for main confounders (Model 3 in Table 4). Specifically, every 300 mg increase in Ca by consuming a portion of dairy products was associated with 24 and 22% lower risk of having high total cholesterol (*p* = 0.003) and high TG levels (*p* = 0.020), respectively. No such associations were observed for Ca from non-dairy sources. No significant associations were observed between Ca from any source and having high LDL-c levels in these models (Table 5).

### 3.4. Associations between Dairy Categories and Lipid Profile

Given these significant associations with Ca from dairy, in the next 3 models, we focused on dairy products of different subtypes (Model 4), fermentation status (Model 5) and fat content (Model 6). The inverse association between dairy Ca intake and high total cholesterol levels was mainly due to milk (*p* = 0.044) and milk-based desserts (*p* = 0.039), i.e., non-fermented (*p* = 0.022), or low-fat dairy products (*p* = 0.005). Every increase in Ca from low-fat dairy products by 300 mg/day was associated with a 32% lower risk of having high total cholesterol. This observation persisted after adjustment for potential confounders [OR 0.72 (0.54, 0.97), *p* = 0.030]. For high TG levels, inverse associations were observed with Ca from milk (43% lower odds, *p* = 0.020) and non-fermented dairy products (40% lower odds, *p* = 0.027), with both associations remaining significant in adjusted models. There were no specific associations between high LDL-c or low HDL-c and dairy Ca by different subtype, fermentation status or fat content. Notably, Ca from high-fat dairy products was not associated with any of the lipid parameters.

### 3.5. Sensitivity Analyses for Calcium from Fermented, High and Low-Fat Dairy Sources

The associations of Ca from fermented, high and low-fat dairy sources with abnormal blood lipids were globally consistent within subgroups in sensitivity analyses (Figure 1 and Figure 2), apart from the following observations. A higher intake of Ca from fermented dairy products was associated with lower prevalence of abnormal cholesterol fractions (high LDL-c, *p* = 0.032; low HDL-c, *p* = 0.025) among individuals taking osteoporosis treatment only. A higher intake of calcium form high-fat dairy products was negatively associated with high TG levels in those without fractures only (*p* = 0.019).

### 3.6. Sensitivity Analyses Using Different Total Cholesterol Thresholds

Given that there is no specific cut-off goal for determining high total cholesterol levels, in sensitivity analyses, we explored the associations between Ca from total and low-fat dairy and high total cholesterol using different cut-offs (7.0, 6.5, 6.0, 5.5 and 5.0 mmol/L). The associations between Ca from total and low-fat dairy and high total cholesterol remained significant for total cholesterol levels ≥6.5 mmol/L (Table 6).

## 4. Discussion

In this cross-sectional analysis of the GERICO cohort, we explored the associations of total calcium intake and calcium from various sources with blood lipids in community-residing elderly. This population is highly relevant because (i) Swiss individuals have a high mean calcium intake, (ii) dairy products are an important part of the Swiss diet, whilst older individuals (iii) are at high risk of musculoskeletal and cardiovascular diseases, (iv) may have misconceptions and (v) receive conflicting advice regarding calcium and dairy product consumption [10,42]. The key findings of our work were that: (i) dairy calcium, and especially calcium from milk and milk-based desserts, i.e., low-fat dairy and non-fermented products were inversely associated with high total cholesterol and/or TG levels, (ii) no significant associations were observed for calcium from non-dairy sources, cheese and high-fat dairy products and abnormal blood lipids, and (iii) increasing calcium intake through supplements was associated with a lower risk of having high TG and low HDL-c levels, although these associations were likely dependent on other factors. Taken together, our results suggest that higher calcium intakes from dietary sources or supplements are not associated with abnormal lipid profiles in the elderly, whilst total and low-fat dairy products are valuable sources of calcium potentially related to favorable blood lipid profiles.

Our study confirms previous findings supporting neutral or protective associations of total or dietary calcium intake with blood lipids [43,44,45,46,47] and provides novel data on calcium from different dietary sources. We showed consistent, inverse associations of dairy calcium with lower odds of having elevated total cholesterol and TG, which were not observed for calcium from non-dairy sources including vegetables, legumes and mineral waters. In line with our results, several cross-sectional studies have shown favorable associations of dairy products with blood lipids [35,48]. Likewise, in a prospective study, an increase of one serving/day in total low-fat dairy consumption was associated with lower increases in total cholesterol, but also LDL-cholesterol levels [49]. In contrast, systematic reviews and/or meta-analyses of RCTs revealed no significant effects of increasing total dairy intake on LDL-c, HDL-c (which is in agreement with the results of our study) and/or TG levels [13,17,50].

Our results further highlight the complex relationship between calcium from different dairy foods and blood lipids. We found inverse associations of calcium from low-fat dairy products (milk, milk-based desserts and fresh cheese) with elevated total cholesterol, while calcium from high-fat dairy products was not associated with abnormal blood lipids. These findings may reflect differences of dairy food matrices in macro- and micronutrients, texture, fat globule size and structure, protein quality and bacterial content, which can impact the bioavailability of contained nutrients and subsequently, their biological effects [12,13,21]. For example, dairy fat, particularly SFAs which have been long thought to increase CVD risk, and whose consumption is restricted by most dietary guidelines, may differentially affect LDL-c depending on the food matrix they ingested (i.e., favorable changes with cheese compared to butter) [14,15,16]. Similarly, epidemiological studies and meta-analyses of RCTs support no or even favorable effects of cheese and/or high-fat dairy products on total cholesterol, LDL-c, HDL-c or TG levels [13,20,35]. These findings suggest that any unfavorable effects of SFAs may be counterbalanced by the beneficial effects of other properties of some dairy matrices. Analogously, the amount and type of fat or other factors contained in high-fat dairy products could have mitigated the potential benefits of calcium on blood lipids, thus contributing to the observed null associations with calcium from high-fat products in our investigation. Further reinforcing the concept of food matrix, some studies support some positive effects of fermented dairy products on cardiometabolic parameters, with these benefits largely attributed to their rich content of probiotics and probiotics’ interactions with the gut microbiome [21]. Our analyses do not confirm these results, because calcium from both fermented and non-fermented products tended to be favorably associated with total cholesterol in univariate models, but these associations were generally attenuated in adjusted models. When we looked at calcium from different fermented products separately (i.e., yogurts, cheese), there were no significant associations with any lipid parameter. The absence of associations in our study may be explained by the overall high intake of calcium in our study population. Indeed, fermented dairy products may improve calcium bioavailability [51]; nevertheless, these effects may be threshold-dependent, and thus less evident in those with higher calcium intakes.

Lastly, we demonstrated that calcium from supplements was associated with a lower prevalence of hypertriglyceridemia and low HDL-c levels in univariate models accounting for total calcium intake; nevertheless, these associations were no longer significant after adjustments, suggesting that they can be largely ascribed to other factors. Two recent meta-analyses of RCTs on calcium supplementation and lipid profile reported that compared to control, calcium supplementation significantly reduced LDL-c levels, whilst no overall effects were seen on total cholesterol or TG levels [29,52]. HDL-c was reported to increase after calcium supplementation in [29], but no effects were found in [52]. Beneficial effects on LDL-c and HDL-c were also reported in a meta-analysis of RCT on this topic in individuals with overweight/obesity [31]. Clinical trials investigating the effects of calcium supplementation on blood lipids, specifically in individuals aged > 50 years, have yielded variable results. For example, decreases in LDL-c and increases in HDL-c were observed after calcium citrate supplementation in older women (mean age of 72 years) [30]. In another RCT, in which middle-aged and older men were randomly assigned to placebo, 600 or 1200 mg Ca/day over a 2-year period, there were no significant treatment effects on serum lipids [32]. Despite discrepancies in study design, supplement types and doses, supplementation duration and populations’ characteristics, our and previous findings support null or favorable associations of calcium supplements and blood lipids overall.

Mechanistically, increasing calcium intake can suppress calcitrophic hormones and reduce adipocyte intracellular Ca^2+^, upregulate lipolysis, inhibit lipogenesis, and lessen lipid storage [27]. In our analysis, despite the fact that our population had overall high calcium intakes, we replicated the finding that higher (dietary and dairy) calcium intakes are associated with lower PTH and β-CTX levels (indicating lower bone resorption). Other possible mechanisms include calcium binding to lipids/bile acids, which may result in their increased fecal excretion, lower amounts of lipids/bile acids entering the enterohepatic circulation and use of hepatic cholesterol for the de novo synthesis of bile acids [27].

Our work has several strengths. It consists of a more comprehensive examination of calcium intake from supplements and various dietary sources including subtypes of dairy products and dairy with different fat content and fermentation status, compared to earlier studies focusing solely on total [44,45] dietary calcium intakes [43,44,45,46,47] or calcium supplements (for reviews see [29,31]). Dietary assessment was performed at multiple timepoints over a ~6-year period, thus our estimates are more representative of habitual dietary intake, account for dietary modifications over the longer term and lessen within-person variability. Important strengths of our statistical approach include the consideration of the total amount of calcium consumed and the adjustments for other sources of calcium in each model, but also for several major confounders including demographic, lifestyle, co-morbidity and medication data.

The limitations of our work should also be acknowledged. Our findings apply to a homogenous cohort of older people with several cardiovascular risk factors and generally high calcium intakes, limiting the extrapolation of the observed associations to more heterogeneous populations or older individuals with lower calcium intakes. We utilized a validated FFQ, which is a retrospective method for dietary assessment sensitive to recall bias. Nevertheless, trained research staff administered the same FFQ at different time points and mean dietary intakes were used; thus, these strategies should have reduced recall bias and variability. Our study design is cross-sectional and although we tried to control for major confounding factors, residual confounders cannot be excluded, thereby precluding causal inferences.

## 5. Conclusions

In conclusion, we observed no or favorable associations between total calcium intake and calcium from different sources, and blood lipid profiles in community-dwelling elderly. Our findings that increasing calcium from total and low-fat dairy products was related to a lower risk of high total cholesterol and/or TG levels, whilst calcium from non-dairy sources and high-fat dairy products was not associated with an increased prevalence of abnormal blood lipids, underline the importance of food matrices in determining health effects. We further observed null (for high total and LDL-cholesterol) or favorable (for high TG and low HDL-c) associations for calcium intake through supplements depending on the lipid parameter assessed. Collectively, these findings contributes to the growing body of evidence on calcium and cardiometabolic health and should be considered when making recommendations of increasing calcium and dairy product consumption for the management of bone fragility in the elderly.

## Figures and Tables

**Figure 1 nutrients-14-01314-f001:**
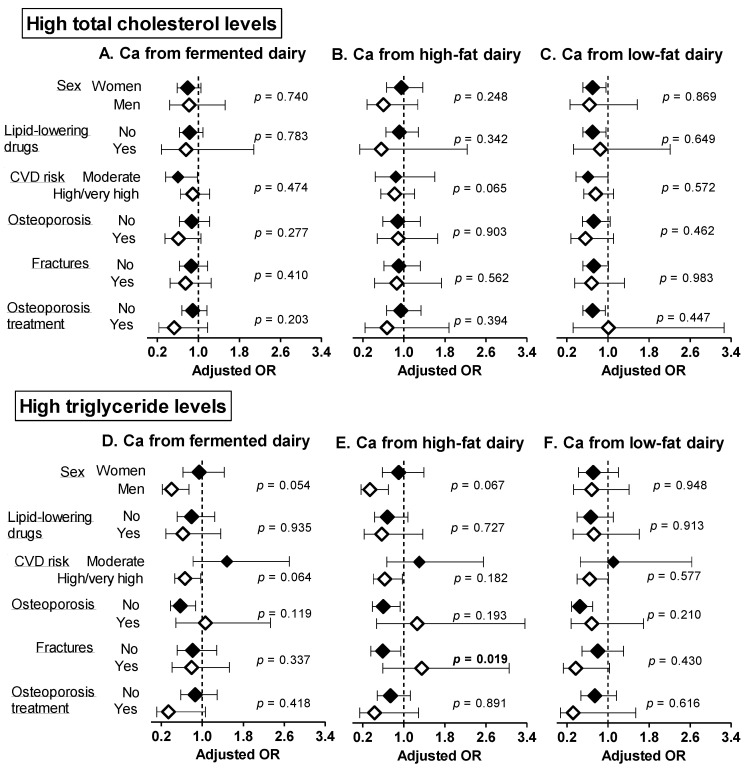
Sensitivity analyses: adjusted ORs and 95 CIs for having high total cholesterol (**A**–**C**) and triglyceride (**D**–**F**) levels per 300 mg increase in calcium intake from fermented, high and low-fat dairy products. The models were adjusted for age, sex, weight, height, smoking status, alcohol intake, dietary energy intake, physical activity energy expenditure and use of medication (statins or other lipid-lowering drugs, antihypertensive drugs, antidiabetic drugs). *p*-values refer to the interaction of each calcium from fermented/high-/low-fat dairy products and the stratifying variable. Significant *p*-values (<0.05) are indicated in bold. Ca: calcium, CVD: cardiovascular disease, TG: triglycerides.

**Figure 2 nutrients-14-01314-f002:**
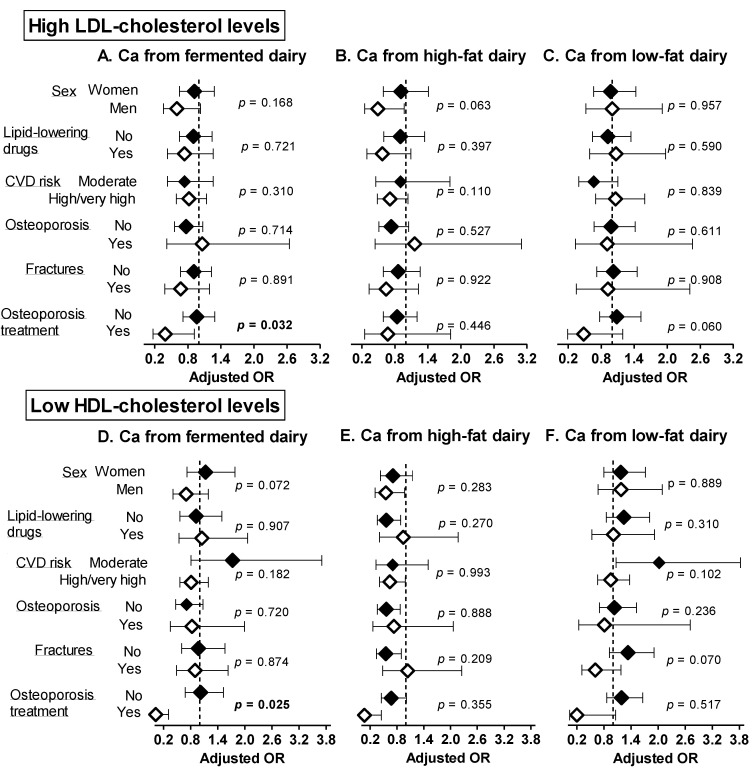
Sensitivity analyses: adjusted ORs and 95 CIs for having high LDL-c (**A**–**C**) and low HDL-c (**D**–**F**) levels per 300 mg increase in calcium intake from fermented, high and low-fat dairy products. Models were adjusted for age, sex, weight, height, smoking status, alcohol intake, dietary energy intake, physical activity energy expenditure and use of medication (statins or other lipid-lowering drugs, antihypertensive drugs, antidiabetic drugs) Results are expressed per 300 mg increase in Ca intake (quantity provided by one serving of dairy products). *p*-values refer to the interaction of each calcium from fermented/high-/low-fat dairy products and the stratifying variable. Significant *p*-values (<0.05) are indicated in bold. Ca: calcium, CVD: cardiovascular disease, HDL-c: high-density lipoprotein cholesterol, LDL-c: low-density lipoprotein cholesterol.

**Table 1 nutrients-14-01314-t001:** Classification of calcium sources (dietary sources & supplements) and their contributions to total calcium intake (in mg/day and %).

Foods/Ca Supplements	Description	Ca (mg/day)	% Total Ca	Total Ca 1527 ± 527 ^1^ (100%) ^2^	Dietary Ca 1189 ± 347 ^1^ (82%) ^2^	Dairy Ca 681 ± 308 ^1^ (46%) ^2^	Fermented Dairy Ca 544 ± 253 ^1^ (37%) ^2^	Non-Fermented Dairy Ca 137 ± 167 ^1^ (9%) ^2^	Low-Fat Dairy Ca 330 ± 225 ^1^ (22%) ^2^	High-Fat Dairy Ca 351 ± 202 ^1^ (24%) ^2^
**Milk**	All types of milk (skimmed, semi-skimmed, whole fat)	116 ± 165	7%	🗸	🗸	🗸		🗸	🗸	
**Yogurts**	All types of yogurts (skimmed, semi-skimmed, whole fat, plain or with fruits)	164 ± 136	11%	🗸	🗸	🗸	🗸		🗸	
**Cheese**	Fresh cheese (e.g., cheese, fromage frais, “petit suisse”)	381 ± 206	26%	🗸	🗸	🗸	🗸		🗸	
	Soft cheese (e.g., Camembert, Tomme, cream cheese, mozzarella)			🗸	🗸	🗸	🗸			🗸
	Hard cheese (e.g., parmesan, gruyere, fondue, raclette, grated cheese)			🗸	🗸	🗸	🗸			🗸
**Milk-based desserts**	Flan, rice pudding, chocolate dessert	21 ± 28	1%	🗸	🗸	🗸		🗸	🗸	
**Non-dairy sources**	Vegetables, legumes, cereals, meat/ fish/poultry and their products, non-dairy beverages, mineral waters	507 ± 141	37%	🗸	🗸					
**Supplements**	Calcium carbonate	338 ± 412	18%	🗸						

^1^ Calcium in mg/d, ^2^ % contribution to total Ca intake.

**Table 2 nutrients-14-01314-t002:** Participants’ characteristics (*n* = 717).

	Total (*n* = 717)
Age (years)	71 ± 2
Sex (women, %)	80
Height (cm)	164 ± 8
Weight (kg)	69 ± 13.5
BMI (kg/m^2^)	25.5 ± 4.4
Obesity (%) ^1^	14
**Blood lipid levels**	
Total cholesterol (mmol/L)	5.74 ± 1.06
LDL-c (mmol/L)	3.37 ± 0.95
HDL-c (mmol/L)	1.83 ± 0.49
TG (mmol/L)	1.21 ± 0.56
**Bone characteristics**	
Spine T-score	−0.91 ± 1.59
Total hip T-score	−0.93 ± 0.95
Femoral neck T-score	−1.47 ± 0.96
Osteoporosis (%) ^2^	21
Prior low-trauma fracture (%)	26
FRAX MOF with BMD (%)	16.3 ± 8.4
Osteoporosis treatment (%) ^3^	15
25-0H vitamin D (nmol/L) ^4^	67.8 ± 27.3
Vitamin D insufficiency (<50 nmol/l) (%) ^4^	29
PTH (pmol/L) ^4^	4.56 ± 1.83
CTX (ng/L) ^4^	380 ± 191
**Dietary and other lifestyle factors**	
Total dietary energy intake (kcal/day)	1515 ± 388
Dietary protein intake (g/kg/day)	1.08 ± 0.29
Total Ca intake (mg/day)	1527 ± 527
Total dairy products (servings/day)	2.8 ± 1.3
Ca supplement users (%)	46
Smoking, current (%)	7
Physical activity energy expenditure (kcal/d)	391 ± 215
Alcohol consumption (≥30 g/day) (%)	10
**Other comorbidities and relevant medications**	
Charlson Comorbidity Index Score	3.0 ± 0.59
Self-reported dyslipidemia or lipid-lowering drugs (%) ^5^	25
Statins or other lipid-lowering drugs (%) ^5^	21
Self-reported diabetes (%) ^5^	5
Anti-diabetic drugs (%) ^5^	5
Self-reported hypertension (%) ^5^	32
Antihypertensive treatment (%) ^5^	30
Self-reported CVD (%) ^5^	6
10-year CVD risk ^6^ Low/moderate (%) High (%) Very high (%)	36 54 10

Values are means ± SDs or percentages. BMD: bone mineral density, Ca: calcium, CTX: β-carboxyterminal cross-linked telopeptide of type I collagen, CVD: cardiovascular disease, HDL-c: high-density lipoprotein, FRAX MOF: 10-year probability of a major osteoporotic fracture, LDL-c: low-density lipoprotein, PTH: parathyroid hormone, TG: triglycerides. ^1^ Obesity defined as a BMI ≥ 30 kg/m^2^. ^2^ Defined as at least one BMD T-score ≤ −2.5 SDs at the lumbar spine, total hip, or femoral neck. ^3^ Bisphosphonates, denosumab, raloxifene, hormone replacement therapy or tibolone. ^4^ As assessed at baseline visit only. ^5^ As reported by participants in a face-to-face interview with a medical doctor. ^6^ Estimated using the Systematic Coronary Risk Estimation 2 and Systematic Coronary Risk Estimation 2-Older Persons risk charts for fatal and non-fatal (myocardial infarction, stroke) CVDs published by the European Society of Cardiology and 12 medical societies. The algorithms take into account the following factors: age, sex, smoking status, systolic blood pressure and non-HDL-c. Participants were classified as being at low-to-moderate (<5% for those aged <70 years, <7.5% for those aged ≥70 years), high (5–10% for those aged <70 years, 7.5–15% for those aged ≥70 years) or very high CVD risk (≥10% for those aged <70 years, ≥15% for those aged ≥70 years).

**Table 3 nutrients-14-01314-t003:** Associations (linear regressions) between PTH and β-CTX (dependent variables) and calcium intakes (continuous independent variable, per 300 mg increase) at the baseline visit of the cohort in the total study population (*n* = 717).

	PTH (pmol/L)				β-CTX (ng/L)			
	Univariate		Adjusted *		Univariate		Adjusted *	
	β (95% CI)	*p*-Value	β (95% CI)	*p*-Value	β (95% CI)	*p*-Value	β (95% CI)	*p*-Value
**Total dietary Ca**	−0.109 (−0.209, −0.008)	**0.034**	−0.088 (−0.186, 0.011)	0.082	−0.010 (−0.020, 0.001)	0.065	−0.010 (−0.021, 0.001)	0.071
**Ca from dairy** **products**	−0.139 (−0.264, −0.014)	**0.029**	−0.112 (−0.233, 0.009)	0.070	−0.012 (−0.024, −0.001)	**0.038**	−0.013 (−0.025, −0.001)	**0.036**
**Ca from non-dairy sources**	−0.015 (−0.283, 0.252)	0.910	−0.007 (−0.270, 0.256)	0.959	−0.001 (−0.030, 0.027)	0.930	0.001 (−0.027, 0.029)	0.943

β-CTX: β-carboxyterminal cross-linked telopeptide of type I collagen, PTH: parathyroid hormone, Ca: Calcium. Results are expressed per 300 mg increase in Ca intake (quantity provided by one serving of dairy products). * Adjusted for age, sex, weight, height, osteoporosis treatment including hormone replacement therapy (HRT), Ca supplement use, 25-0H vitamin D and creatinine levels. Significant *p*-values (< 0.05) are indicated in bold.

**Table 4 nutrients-14-01314-t004:** Univariate and adjusted ORs and 95 CIs for having high total cholesterol and high triglyceride levels per 300 mg increase in total calcium intake and calcium from different sources.

	High Total Cholesterol (≥6.5 mmol/L)				High Triglycerides (≥1.7 mmol/L)			
	Univariate		Adjusted *		Univariate		Adjusted *	
	OR (95%)	*p*-Value	OR (95%)	*p*-Value	OR (95%)	*p*-Value	OR (95%)	*p*-Value
**Model 1**								
Total Ca intake	0.90 (0.81, 0.99)	**0.038**	0.89 (0.79, 1.01)	0.071	0.83 (0.73, 0.95)	**0.007**	0.85 (0.73, 1)	0.054
**Model 2**								
Dietary Ca	0.83 (0.71, 0.96)	**0.014**	0.85 (0.70, 1.03)	0.091	0.85 (0.71, 1.02)	0.081	0.77 (0.59, 1.02)	0.066
Ca from supplements	0.95 (0.83, 1.08)	0.440	0.91 (0.79, 1.06)	0.235	0.82 (0.69, 0.97)	**0.022**	0.90 (0.75, 1.08)	0.250
**Model 3**								
Ca from dairy products	0.76 (0.63, 0.91)	**0.003**	0.79 (0.64, 0.98)	**0.034**	0.78 (0.63, 0.96)	**0.020**	0.71 (0.53, 0.96)	**0.025**
Ca from non-dairy sources	1.16 (0.77, 1.73)	0.482	1.15 (0.75, 1.76)	0.520	1.17 (0.77, 1.79)	0.456	1.12 (0.69, 1.81)	0.647
Ca from supplements	0.96 (0.84, 1.09)	0.509	0.92 (0.79, 1.07)	0.280	0.83 (0.7, 0.98)	**0.027**	0.90 (0.75, 1.09)	0.283
**Model 4**								
Ca from milk	0.68 (0.47, 0.99)	**0.044**	0.71 (0.48, 1.06)	0.093	0.57 (0.35, 0.91)	**0.020**	0.52 (0.31, 0.87)	**0.014**
Ca from yogurts	0.78 (0.51, 1.19)	0.250	0.79 (0.49, 1.28)	0.343	0.84 (0.42, 1.69)	0.633	0.92 (0.45, 1.87)	0.811
Ca from cheese	0.84 (0.65, 1.08)	0.173	0.85 (0.62, 1.16)	0.295	0.85 (0.64, 1.13)	0.266	0.74 (0.52, 1.06)	0.103
Ca from milk-based desserts	0.10 (0.01, 0.89)	**0.039**	0.17 (0.02, 1.63)	0.124	2.16 (0.27, 17.64)	0.471	0.68 (0.05, 9.21)	0.768
Ca from non-dairy sources	1.16 (0.77, 1.74)	0.491	1.14 (0.74, 1.77)	0.553	1.14 (0.74, 1.75)	0.544	1.11 (0.69, 1.80)	0.659
Ca from supplements	0.96 (0.84, 1.1)	0.553	0.92 (0.79, 1.07)	0.293	0.82 (0.7, 0.98)	**0.027**	0.91 (0.75, 1.09)	0.298
**Model 5**								
Ca from fermented dairy	0.81 (0.65, 1.00)	0.054	0.84 (0.65, 1.08)	0.176	0.86 (0.66, 1.12)	0.260	0.80 (0.55, 1.15)	0.222
Ca from non-fermented dairy	0.64 (0.44, 0.94)	**0.022**	0.70 (0.47, 1.04)	0.076	0.60 (0.38, 0.94)	**0.027**	0.53 (0.32, 0.89)	**0.017**
Ca from non-dairy sources	1.14 (0.76, 1.71)	0.529	1.15 (0.75, 1.76)	0.529	1.15 (0.75, 1.77)	0.517	1.12 (0.69, 1.82)	0.643
Ca from supplements	0.96 (0.84, 1.09)	0.513	0.92 (0.79, 1.07)	0.281	0.83 (0.7, 0.98)	**0.028**	0.91 (0.76, 1.09)	0.297
**Model 6**								
Ca from high-fat dairy	0.85 (0.65, 1.11)	0.240	0.89 (0.65, 1.23)	0.496	0.84 (0.63, 1.13)	0.242	0.73 (0.50, 1.05)	0.091
Ca from low-fat dairy	0.68 (0.52, 0.89)	**0.005**	0.72 (0.54, 0.97)	**0.030**	0.73 (0.51, 1.06)	0.097	0.70 (0.45, 1.07)	0.102
Ca from non-dairy sources	1.15 (0.76, 1.72)	0.508	1.16 (0.75, 1.78)	0.504	1.17 (0.77, 1.79)	0.466	1.12 (0.69, 1.81)	0.640
Ca from supplements	0.96 (0.84, 1.10)	0.543	0.92 (0.79, 1.07)	0.279	0.83 (0.7, 0.98)	**0.029**	0.90 (0.75, 1.09)	0.286

* Adjusted for age, sex, weight, height, smoking status, alcohol intake, dietary energy intake, physical activity energy expenditure and use of medication (statins or other lipid-lowering drugs, antihypertensive drugs, antidiabetic drugs). Results are expressed per 300 mg increase in Ca intake (quantity provided by one serving of dairy products). Significant *p*-values (<0.05) are indicated in bold. All models consider the total amount of Ca consumed. Model 1: Total Ca intake, Model 2: Ca from diet and supplements, Model 3: Ca from dairy and non-dairy sources, Model 4: Ca from dairy sources by subtype, Model 5: Ca from dairy sources by fermentation status, Model 6: Ca from dairy sources by fat content.

**Table 5 nutrients-14-01314-t005:** Univariate and adjusted ORs and 95 CIs for having high LDL-c and low HDL-c levels per 300 mg increase in total calcium intake and calcium from different sources.

	High LDL-c (≥2.6 mmol/L)				Low HDL-c (<1.3 mmol/L)			
	Univariate		Adjusted *		Univariate		Adjusted *	
	OR (95%)	*p*-Value	OR (95%)	*p*-Value	OR (95%)	*p*-Value	OR (95%)	*p*-Value
**Model 1**								
Total Ca intake	1.01 (0.91, 1.12)	0.805	0.95 (0.84, 1.07)	0.401	0.85 (0.75, 0.96)	**0.010**	0.89 (0.76, 1.05)	0.164
**Model 2**								
Dietary Ca	0.94 (0.81, 1.1)	0.454	0.89 (0.71, 1.13)	0.343	1.04 (0.88, 1.23)	0.612	0.91 (0.70, 1.18)	0.469
Ca from supplements	1.07 (0.93, 1.23)	0.358	0.98 (0.84, 1.14)	0.769	0.71 (0.58, 0.86)	**0.001**	0.88 (0.72, 1.08)	0.230
**Model 3**								
Ca from dairy products	0.93 (0.78, 1.12)	0.443	0.90 (0.69, 1.15)	0.396	1.07 (0.88, 1.29)	0.490	0.93 (0.70, 1.24)	0.622
Ca from non-dairy sources	0.99 (0.69, 1.44)	0.978	0.88 (0.53, 1.44)	0.602	0.94 (0.63, 1.39)	0.739	0.79 (0.48, 1.31)	0.362
Ca from supplements	1.07 (0.93, 1.23)	0.353	0.98 (0.84, 1.14)	0.766	0.71 (0.58, 0.86)	**0.001**	0.88 (0.72, 1.08)	0.226
**Model 4**								
Ca from milk	0.96 (0.70, 1.32)	0.794	0.99 (0.64, 1.53)	0.953	1.14 (0.81, 1.62)	0.447	0.88 (0.58, 1.34)	0.540
Ca from yogurts	0.91 (0.61, 1.35)	0.624	0.88 (0.57, 1.36)	0.557	1.19 (0.73, 1.93)	0.492	1.55 (0.96, 2.50)	0.070
Ca from cheese	0.91 (0.69, 1.19)	0.493	0.83 (0.60, 1.16)	0.277	0.93 (0.7, 1.24)	0.614	0.68 (0.47, 0.99)	**0.047**
Ca from milk-based desserts	2.28 (0.3, 17.45)	0.428	4.52(0.42, 48.12)	0.211	5.29 (0.76, 36.85)	0.092	1.05 (0.06, 17.3)	0.970
Ca from non-dairy sources	0.99 (0.68, 1.44)	0.964	0.88 (0.53, 1.44)	0.608	0.93 (0.62, 1.38)	0.712	0.75 (0.45, 1.24)	0.263
Ca from supplements	1.07 (0.93, 1.23)	0.377	0.97 (0.84, 1.14)	0.747	0.70 (0.57, 0.85)	**<0.001**	0.87 (0.71, 1.07)	0.199
**Model 5**								
Ca from fermented dairy	0.91 (0.73, 1.14)	0.414	0.84 (0.65, 1.10)	0.214	1.02 (0.79, 1.3)	0.901	0.93 (0.64, 1.36)	0.704
Ca from non-fermented dairy	0.98 (0.71, 1.34)	0.893	1.01 (0.65, 1.57)	0.971	1.19 (0.85, 1.67)	0.323	0.93 (0.63, 1.39)	0.736
Ca from non-dairy sources	1.00 (0.69, 1.45)	0.999	0.88 (0.53, 1.45)	0.612	0.95 (0.64, 1.4)	0.783	0.79 (0.48, 1.31)	0.362
Ca from supplements	1.07 (0.93, 1.23)	0.353	0.98 (0.84, 1.14)	0.778	0.71 (0.58, 0.86)	**<0.001**	0.88 (0.72, 1.08)	0.225
**Model 6**								
Ca from high-fat dairy	0.88 (0.67, 1.16)	0.365	0.78 (0.56, 1.07)	0.126	0.91 (0.68, 1.22)	0.540	0.65 (0.45, 0.95)	**0.025**
Ca from low-fat dairy	0.97 (0.77, 1.23)	0.833	0.99 (0.72, 1.35)	0.936	1.20 (0.93, 1.55)	0.168	1.14 (0.83, 1.57)	0.416
Ca from non-dairy sources	1.00 (0.69, 1.45)	0.985	0.86 (0.52, 1.41)	0.550	0.94 (0.63, 1.38)	0.739	0.75 (0.45, 1.25)	0.273
Ca from supplements	1.07 (0.93, 1.23)	0.365	0.98 (0.84, 1.14)	0.776	0.70 (0.58, 0.85)	**<0.001**	0.87 (0.71, 1.07)	0.191

* Adjusted for age, sex, weight, height, smoking status, alcohol intake, dietary energy intake, physical activity energy expenditure and use of medication (statins or other lipid-lowering drugs, antihypertensive drugs, antidiabetic drugs). Results are expressed per 300 mg increase in Ca intake (quantity provided by one serving of dairy products). Significant *p*-values (< 0.05) are indicated in bold. All models consider the total amount of Ca consumed. Model 1: Total Ca intake, Model 2: Ca from diet and supplements, Model 3: Ca from dairy and non-dairy sources, Model 4: Ca from dairy sources by subtype, Model 5: Ca from dairy sources by fermentation status, Model 6: Ca from dairy sources by fat content.

**Table 6 nutrients-14-01314-t006:** Univariate and adjusted ORs and 95 CIs for having high total cholesterol according to various cut-off values per 300 mg increase in total calcium intake total and low-fat dairy products.

High Total Cholesterol Cut-offs	Ca from Dairy Products				Ca from Low-Fat Dairy Products			
	Univariate		Adjusted *		Unadjusted		Adjusted *	
	OR (95% CI)	*p*-Value	OR (95% CI)	*p*-Value	OR (95% CI)	*p*-Value	OR (95% CI)	*p*-Value
**≥7.0 mmol/L**	0.78 (0.62, 0.99)	**0.037**	0.79 (0.60, 1.04)	0.088	0.66 (0.45, 0.96)	**0.032**	0.69 (0.46, 1.03)	0.070
**≥6.5 mmol/L**	0.76 (0.63, 0.91)	**0.003**	0.79 (0.64, 0.98)	**0.034**	0.68 (0.52, 0.89)	**0.005**	0.72 (0.54, 0.97)	**0.030**
**≥6.0 mmol/L**	0.88 (0.76, 1.01)	0.076	0.89 (0.73, 1.07)	0.215	0.84 (0.69, 1.03)	0.099	0.86 (0.67, 1.10)	0.220
**≥5.5 mmol/L**	0.91 (0.78, 1.05)	0.197	0.89 (0.72, 1.09)	0.268	0.90 (0.74, 1.10)	0.315	0.91 (0.71, 1.18)	0.4830
**≥5.0 mmol/L**	1.00 (0.83, 1.20)	0.972	1.03 (0.80, 1.33)	0.800	1.01 (0.80, 1.27)	0.954	1.05 (0.78, 1.40)	0.7680

* Adjusted for age, sex, weight, height, smoking status, alcohol intake, dietary energy intake, physical activity energy expenditure and use of medication (statins or other lipid-lowering drugs, antihypertensive drugs, antidiabetic drugs) Results are expressed per 300 mg increase in Ca intake (quantity provided by one serving of dairy products). Significant *p*-values (*p* < 0.05) are indicated in bold. All models consider the total amount of Ca consumed.

## Data Availability

The data presented in this study are available on request from the corresponding author if proper IRB approval and material transfer agreements are obtained. Due to restrictions based on privacy regulations and informed consent of the participants, data cannot be made freely available in a public repository.

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
