# Peer review of "Associations of Calcium Intake and Calcium from Various Sources with Blood Lipids in a Population of Older Women and Men with High Calcium Intake"

_nutrients, 2022, doi:10.3390/nu14061314_

Round 1
Reviewer 1 Report
This cross-sectional study assessed the relation of ca intake from different sources with lipid profile in elderly people. Generally, the study is well-designed and the manuscript is well-written, but some minor issues should be addressed as follows:
Abstract have no methods.
Please mention the number of FFQ’s food items.
Table 1 is vague. What did authors tend to present in it?
What is the relevance between table 3 and the manuscript aims? It seems irrelevant.
There are 2 meta-analysis on the effects of calcium and dairy on lipid profile which are recommended to be used in discussion section.
No adverse effects of dairy products on lipid profile: A systematic review and meta-analysis of randomized controlled clinical trials.
Derakhshandeh-Rishehri SM, Ghobadi S, Akhlaghi M, Faghih S. Diabetes Metab Syndr. 2021 Nov-Dec;15(6):102279. doi: 10.1016/j.dsx.2021.102279. Epub 2021 Sep 13
The effect of calcium supplement intake on lipid profile: a systematic review and meta-analysis of randomized controlled clinical trials.
Derakhshandeh-Rishehri SM, Ghobadi S, Akhlaghi M, Faghih S. Crit Rev Food Sci Nutr. 2020 Nov 23:1-10. doi: 10.1080/10408398.2020.1850414. Online ahead of print. PMID: 33226265
Author Response
This cross-sectional study assessed the relation of ca intake from different sources with lipid profile in elderly people. Generally, the study is well-designed and the manuscript is well-written, but some minor issues should be addressed as follows:
We thank the reviewer for his/her positive comments.
Abstract have no methods.
We have now added a description of our methods to the abstract.
Revised manuscript, page 1: “Dietary calcium intake was assessed at several timepoints using a validated FFQ and calcium supplement use was recorded. Blood lipids were treated as categorical variables to distinguish those with normal and abnormal levels”.
Please mention the number of FFQ’s food items.
The FFQ contained 25 items. We have now added this information in the text.
Revised manuscript, page 3: “A validated food-frequency questionnaire (FFQ) containing 25 items and designed to assess calcium and protein intakes was used to assess habitual dietary intake including dairy products consumption over the preceding year at all study visits (adapted from [34])”.
Table 1 is vague. What did authors tend to present in it?
Table 1 aimed to provide a graphical representation of the study exposure (total calcium intake and calcium sources i.e., which foods/food items were included in each category), alongside a detailed investigation of these dietary parameters in our population. It is envisaged that the data included in Table 1 will help readers interpret the results of our models (Models 1-6).
What is the relevance between table 3 and the manuscript aims? It seems irrelevant.
Our results regarding calcium intake and lipid profile are observed in a population in whom the benefits of calcium intake on bone-related outcomes are also demonstrated. Specifically, despite the fact that our population had a high calcium intake, we reproduced the findings that dietary calcium intake (in particular Ca from dairy products) was associated with lower PTH and β-CTX (a marker of bone resorption) levels. Replicating these well-established associations validates our calcium intake dietary assessments. Finally, Table 3 is relevant to this work as our results for calcium intake and lipid profile have implications when making recommendations of increasing calcium and dairy products consumption for the management of bone fragility in elderly. If the reviewer/editor still think that Table 3 is less relevant to this work, we propose to move this table as supplementary material.
There are 2 meta-analysis on the effects of calcium and dairy on lipid profile which are recommended to be used in discussion section.
No adverse effects of dairy products on lipid profile: A systematic review and meta-analysis of randomized controlled clinical trials.
Derakhshandeh-Rishehri SM, Ghobadi S, Akhlaghi M, Faghih S. Diabetes Metab Syndr. 2021 Nov-Dec;15(6):102279. doi: 10.1016/j.dsx.2021.102279. Epub 2021 Sep 13
The effect of calcium supplement intake on lipid profile: a systematic review and meta-analysis of randomized controlled clinical trials.
Derakhshandeh-Rishehri SM, Ghobadi S, Akhlaghi M, Faghih S. Crit Rev Food Sci Nutr. 2020 Nov 23:1-10. doi: 10.1080/10408398.2020.1850414. Online ahead of print. PMID: 33226265
We would like to thank the reviewer for these suggestions, we have now included both references in our manuscript (ref 50, 52: in pages 15 -16)
Reviewer 2 Report
The authors investigated the association between calcium uptake and cardiometabolic health in elderly. They found that high calcium from dairy products were associated with lower risk of high total cholesterol and / or TG levels.
Comments
- Lines 216-239: the authors should not repeat in the text the data presented in the Table 2. This should be corrected.
- Lines 240-242: This sentence is not clear it should be rephrased.
- Discussion, Lines 364-366: “This population is highly relevant since i) Swiss individuals have a high mean calcium intake, …. iii) are at high risk for musculoskeletal and cardiovascular diseases,…” These statements contradict with authors findings. This should be corrected.
- Lines 387-390: The importance of the present study is not clear as it does not reveal the reason of the discrepancy with the results of previous studies [13,17]. This should be addressed.
- Lines 413-415: The authors should provide explanation why their results do not confirm results of others [21].
- Table 2: The authors should stratify their patients according to their health status indicating the diseases burden in different subsets, CVD in particular.
Author Response
The authors investigated the association between calcium uptake and cardiometabolic health in elderly. They found that high calcium from dairy products were associated with lower risk of high total cholesterol and / or TG levels.
Lines 216-239: the authors should not repeat in the text the data presented in the Table 2. This should be corrected.
We agree with the reviewer, and we have now removed some data mentioned in the text, see revised manuscript (revised manuscript, page 7)
Lines 240-242: This sentence is not clear it should be rephrased.
We agree with the reviewer, this sentence has now be revised and reads as follows:
Revised manuscript, page 8: “Individuals at low/moderate CVD risk had a higher total Ca intake compared to those at high/very high risk (Ca intake: 1615± 522 mg/day vs. 1477±523 mg/day, respectively; P < 0.001)”.
Discussion, Lines 364-366: “This population is highly relevant since i) Swiss individuals have a high mean calcium intake, …. iii) are at high risk for musculoskeletal and cardiovascular diseases,…” These statements contradict with authors findings. This should be corrected.
It is not clear to us how these statements contradict to our findings.
Statement i: “Swiss individuals have a high mean calcium intake”
Mean total Ca intake was 1527 ± 527 mg/day. Most of the participants (84%) were meeting Ca recommendations in Switzerland (1000 mg/d) and only 4% of them were consuming <700 mg Ca/day.
Statement iii: “are at high risk for musculoskeletal and cardiovascular diseases”
In our investigation, one in 5 participants had osteoporosis and one in 4 participants had experienced low-trauma fractures. Furthermore, based on SCORE2/SCORE2-OP charts, 64% of the participants were at high or very high 10-year CVD risk.
Lines 387-390: The importance of the present study is not clear as it does not reveal the reason of the discrepancy with the results of previous studies [13,17]. This should be addressed.
In fact our results do not contradict ref 13, 17. In line with these recent meta-analyses, we found no associations between Ca from total dairy intake and abnormal sub-fractions of cholesterol (low HDL-c levels and/or high LDL-c levels). To enhance clarity, we have rephrased this part of the text, which now reads:
Revised manuscript, page 15: “In contrast, systematic reviews and/or meta-analyses of RCTs revealed no significant effects of increasing total dairy intake on LDL-c, HDL-c (which is in agreement with the results of our study) and/or TG levels [13,17,50]”.
Lines 413-415: The authors should provide explanation why their results do not confirm results of others [21].
We would like to thank the reviewer for this comment. In our work, calcium from both fermented and non-fermented products tended to be favorably associated with total cholesterol in univariate models, but these associations were generally attenuated in adjusted models. The absence of favorable associations of fermented products with lipid profile in adjusted models may be due to the higher Ca intakes of participants in the present study. Indeed, the benefit of fermented dairy products, known to increase calcium bioavailability though various mechanisms, may be attenuated in case of high calcium intake, as a result of threshold effect (i.e., calcium absorption also increases with the amount of calcium). We have now added a relevant description to the discussion:
Revised manuscript, page 15: “The absence of our associations in our study may be explained by the overall high intake of our study population. Indeed, fermented dairy products may improve calcium bioavailability [51], nevertheless, these effects may be threshold dependent and thus, less evident in those with higher calcium intakes”.
Table 2: The authors should stratify their patients according to their health status indicating the diseases burden in different subsets, CVD in particular.
We are not sure that we understand the comment of the reviewer. Since only 6% (n = 45) of the participants in this work had experienced a CVD event, we believe that stratification of participants according to this factor would lead to very unbalanced groups. If we stratify participants according to CVD risk (as assessed by SCORE2/SCORE2-OP charts), several characteristics described in Table 2 are already considered in the calculation of the algorithm (i.e., sex, age, smoking, LDL-levels), and as such, any differences in these parameters would be expected.